# The Molecular Mechanism of Farnesoid X Receptor Alleviating Glucose Intolerance in Turbot (*Scophthalmus maximus*)

**DOI:** 10.3390/cells13231949

**Published:** 2024-11-23

**Authors:** Gaochan Qin, Mingzhu Pan, Dong Huang, Xinxin Li, Yue Liu, Xiaojun Yu, Kangsen Mai, Wenbing Zhang

**Affiliations:** 1The Key Laboratory of Aquaculture Nutrition and Feeds (Ministry of Agriculture and Rural Affairs), The Key Laboratory of Mariculture (Ministry of Education), Fisheries College, Ocean University of China, Qingdao 266003, China; ginny0315@stu.ouc.edu.cn (G.Q.);; 2College of Marine and Biology Engineering, Yancheng Institute of Technology, Yancheng 224051, China; mingzhupan@ycit.edu.cn

**Keywords:** transcriptional activity of FXR, amino acid sequences, gene expression, gluconeogenesis, primary hepatocyte

## Abstract

To explore the molecular targets for regulating glucose metabolism in carnivorous fish, the turbot (*Scophthalmus maximus*) was selected as the research object to study. Farnesoid X receptor (FXR; NR1H4), as a ligand-activated transcription factor, plays an important role in glucose metabolism in mammals. However, the mechanisms controlling glucose metabolism mediated by FXR in fish are not understood. It was first found that the protein levels of FXR and its target gene, small heterodimer partner (SHP), were significantly decreased in the high-glucose group (50 mM, HG) compared with those in the normal glucose group (15 mM, CON) in primary hepatocytes of turbot. By further exploring the function of FXR in turbot, the full length of FXR in turbot was cloned, and its nuclear localization function was characterized by subcellular localization. The results revealed that the FXR had the highest expression in the liver, and its capability to activate SHP expression through heterodimer formation with retinoid X receptor (RXR) was proved, which proved RXR could bind to 15 binding sites of FXR by forming hydrogen bonds. Activation of FXR in both the CON and HG groups significantly increased the expression of glucokinase (*gk*) and pyruvate kinase (*pk*), while it decreased the expression of cytosolic phosphoenolpyruvate carboxykinase (*cpepck*), mitochondrial phosphoenolpyruvate carboxykinase (*mpepck*), glucose-6-phosphatase 1 (*g6pase1*) and glucose-6-phosphatase 2 (*g6pase2*), and caused no significant different in glycogen synthetase (*gs*). ELISA experiments further demonstrated that under the condition of high glucose with activated FXR, it could significantly decrease the activity of PEPCK and G6PASE in hepatocytes. In a dual-luciferase reporter assay, the FXR could significantly inhibit the activity of G6PASE2 and cPEPCK promoters by binding to the binding site ‘ATGACCT’. Knockdown of SHP after activation of FXR reduced the inhibitory effect on gluconeogenesis. In summary, FXR can bind to the *mpepck* and *g6pase2* promoters to inhibit their expression, thereby directly inhibiting the gluconeogenesis pathway. FXR can also indirectly inhibit the gluconeogenesis pathway by activating *shp*. These findings suggest the possibility of FXR as a molecular target to regulate glucose homeostasis in turbot.

## 1. Introduction

Compared to mammals, fish, especially carnivorous fish, can effectively use non-sugar precursors (e.g., amino acids and fatty acids) to synthesize glucose. Therefore, they do not need to take more extra carbohydrates [1,2]. Even under fasting or starvation conditions, the proportion of energy provided by protein and lipid decomposition in fish exceeds that of glycogen hydrolysis [3]. Furthermore, excessive carbohydrates in the diet can lead to abnormal depositions of glycogen and lipids in the liver of fish [4,5]. There are different opinions on the hypothesis for why carnivorous fish cannot make full use of carbohydrates, e.g., low glycolysis efficiency, poor glycogen and lipid synthesis, and lack of glucose transporters. The more controversial conjecture is that the hepatic gluconeogenesis pathway is not effectively inhibited [4,6,7]. Gluconeogenesis, as a mechanism in response to changes in nutrition and hormones during the regulation of glucose homeostasis, can be activated under external stimulation to produce glucose [8]. The inhibition of the crucial genes in the gluconeogenesis pathway (e.g., glucose-6-phosphatase, *g6pase*, and phosphoenolpyruvate carboxykinase, *pepck*) after a high-carbohydrate diet intake is one of the important processes for mammals to regulate glucose homeostasis. However, it was found in carnivorous fish that the gluconeogenesis pathway was not inhibited like that in mammals after increasing dietary carbohydrate levels, and endogenous glucose was even still produced uncontrolled in some species [1]. The mRNA level and activity of the gluconeogenesis key gene *g6pase* were not affected by dietary carbohydrates in a study of rainbow trout and sea bream (*Sparus aurata*) [9,10]. Similarly, a high-carbohydrate diet significantly increased the blood glucose level and the expression of *pepck1* in Japanese flounder, *Paralichthys olivaceus* [7]. Therefore, the search for molecular targets and pathways for the regulation of glucose metabolism in carnivorous fish is a research hotspot. In the diabetic mouse model, it was found that specific activation of farnesoid X receptor (FXR) could significantly inhibit the expression of key gluconeogenesis genes *pepck* and *g6pase*, thereby reducing blood glucose levels [11]. The expression of gluconeogenesis genes decreased significantly with the activation of FXR in mice and HepG2 cells [12,13]. FXR can be activated by specific ligands to selectively bind to the promoter of downstream target genes and regulates the transcription of related genes. Similar hormone and enzyme systems were found in fish and mammals, so it is reasonable to speculate that FXR can be used as a molecular target to regulate gluconeogenesis and alleviate hyperglycemia in carnivorous fish.

FXR is a member of the nuclear receptor superfamily, and is a ligand-activated transcription factor that can be activated by specific bile acids, such as chenodeoxycholic acid (CDCA) [14,15]. FXR regulates the expression of downstream genes in the form of a monomer or heterodimer with retinoid X receptor (RXR), and is involved in the regulation of bile acid, glucose, lipids and inflammatory responses [16]. Among these, FXR has been shown to be involved in glucose metabolism by regulating liver gluconeogenesis mainly through the target gene small heterodimer partner (SHP) in mice [17]. Dysfunction of FXR causes severe liver diseases such as cholestasis, liver infection, cirrhosis and hepatocellular carcinoma [18]. FXR in mice inhibited the endoplasmic reticulum stress-induced inflammasome in hepatocytes to ameliorate liver injury [19]. After using bile acids to activate FXR in mice, FXR could bind to the specific FXR response elements (FXRE) in PEPCK and G6PASE promoters to regulate their expression negatively [17]. According to statistics, typical FXREs are composed of the typical AGGTCA hexanucleotide core motif, with an interval of 0 bp (IR-0) or 1 bp (IR-1) [20,21]. Several studies have been carried out to demonstrate the effects of activation of FXR on the lipid metabolism in large yellow croaker (*Larimichthys crocea*) [22] and the inflammatory response in zebrafish (*Danio rerio*) [23] and rainbow trout (*Oncorhynchus mykiss*) [24]. As a specific agonist of FXR synthesized in vitro, GW4064 has been proven to significantly activate FXR in the liver of Japanese medaka (*Oryzias latipes*) [25]. However, no published studies on the function of FXR in the glucose metabolism in fish were available.

Turbot (*Scophthalmus maximus*), as a carnivorous fish, has been proven to be glucose-intolerant [26]. In our previous study, it was found that feeding diets with high carbohydrates could lead to abnormal glucose metabolism, which is manifested by a continuous increase in blood glucose in turbot. Furthermore, the expression of *g6pase2* in the liver increased significantly with high dietary carbohydrates [27]. Therefore, the purpose of the present study is to explore whether FXR can be used as a molecular target to regulate glucose metabolism in turbot, which provides the basal data to clarify the causes of glucose intolerance in turbot.

## 2. Materials and Methods

All the experiments and procedures in the present study were carried out in strict accordance with the guidelines for the Use of Experimental Animals of Ocean University of China.

### 2.1. Animals and Sampling

All turbots (body weight: 20.0 ± 1.8 g) were purchased from the Lingyue Aquaculture Farm (Rizhao, Shandong Province, China) and cultured in a re-circulating seawater system in the laboratory with a water temperature of 18 ± 2 °C. The turbots were fed with commercial feed twice a day (8:00 a.m. and 17:00 p.m.) for subsequent experimental sampling and cell culture. Six turbots were randomly selected for tissue sampling. Twelve kinds of tissues, including skin, liver, intestine, muscle, gill, brain, spleen, stomach, adipose, head kidney, eye and heart tissues, were isolated for subsequent detection of *fxr* expression in different tissues. All samples were stored at −80 °C until analysis.

### 2.2. Cloning and Analysis of fxr Gene

Total mRNA in the liver was extracted with Trizol Reagent (Takara, Shiga, Japan) and electrophoresed on a 1.2% denaturing agarose gel to test its integrity. The quality and concentration of RNA were measured with Nanodrop 2000 (Thermo Fisher Scientific, Wilmington, DE, USA). Subsequently, the obtained RNA was transcribed into cDNA with the reverse transcription kit (Accurate Biology, Changsha, China) of Moloney Murine Leukemia Virus (M-MLV) reverse transcription polymerase. The sequence of *fxr* was obtained from the genome data of turbot (Taxonomy ID: 52904) in the National Center for Biotechnology Information (NCBI). After that, the amplification primers (Appendix A) were designed with DNAMAN 8 software for gene cloning and synthesized by Sangon Biotech (https://store.sangon.com/newPrimer) accessed on 7 June 2022. DNAMAN 8 and MEGA 8.0 software were used for sequence analysis, and the SWILL-MODLE workspace (https://swissmodel.expasy.org/) accessed on 5 December 2022. was used for protein structure model prediction.

### 2.3. Cell Culture and Treatment

#### 2.3.1. Cell Culture

The turbot was temporarily cultured in seawater containing 0.1% penicillin and streptomycin (Hyclone, Logan, UT, USA) for 24 h. After anesthesia with eugenol (1:10,000) (Macklin, Shanghai, China), the turbot was disinfected with 75% alcohol, and blood was taken from the caudal vertebrae. The liver was separated with a sterilized dissector and washed in cold phosphate-buffered saline (PBS) (Biosharp, Hefei, China) containing 1% penicillin and streptomycin to remove impurities. Then, the liver was cut into small pieces with a size of about 1 mm^3^ in Eagle’s medium (DMEM)–F12 medium (Gibco, New York, USA). The small pieces of liver tissue were transferred to a 15 mL centrifuge tube, supplemented with 0.25% trypsin–EDTA (Gibco, New York, NY, USA), and placed on a Vertical Mixer (QQMV-100, Shanghai, China) for low-speed shaking digestion for 20 min. During digestion, the supernatant was collected every 5 min and the same amount of complete culture medium containing DMEMF12, 15% fetal bovine serum (FBS; Gibco, New York, NY, USA), and 0.1% penicillin and streptomycin was added to the supernatant. The digested liquid was sieved through 200 mesh (70 μm), and centrifuged at 1000 rpm for 3 min, and the obtained cell precipitate was resuspended with the complete culture medium and inoculated into cell culture plates. The cell culture bottles were placed in a biochemical incubator (SPX-100B-Z, Shanghai, China) at 23 °C to start the primary cell culture. After 2–3 passages, subsequent experiments could be processed.

#### 2.3.2. Different Glucose Concentration Treatments

Glucose-free DMEM (Solarbio, Beijing, China) containing 15% FBS was prepared, and glucose was added to make the final concentrations of glucose in the medium 15 mM (control glucose, CON) and 50 mM (high glucose, HG), and then the medium was filtered with a 0.22 μm filter membrane.

#### 2.3.3. Activator Treatment

Cells were seeded in 96-well plates or 6-well plates with a density of 1 × 10^6^ cells/mL, 100 μL per well in 96-well plates and 2 mL per well in 6-well plates. Different concentrations of FXR agonists were prepared: GW4064 (0, 0.5, 1, 2, 4 and 8 μM) and CDCA (0, 25, 50, 100, 150 and 200 μM). After 24 h of treatment, cells in 96-well plates were used for cell viability, and cells in 96-well plates were collected for gene expression detection. Different time points (0, 12, 24 and 48 h) with selected agonist concentrations were analyzed, and the time and concentration for the activator treatment were finally determined.

#### 2.3.4. Gene Overexpression and Knockdown

The pcDNA3.1-SHP plasmid was constructed and the quality and concentration of the effort were detected by Nanodrop 2000. Three pairs of siSHP (siSHP-171, siSHP-495 and siSHP-532) and siSHP-NC were designed and chemically synthesized at Shanghai Sangon Biotech Co., Ltd. Three groups of siSHP were used as the experimental group, NC as the control group and untreated cells as the control group. The specific sequence is shown in Appendix A. Both overexpression and knockdown were treated with the universal transfection reagent (YEASEN, Shanghai, China).

### 2.4. Transcription Activity in Yeast

To determine the transcription activity of FXR, a yeast transcription assay kit (MH102, Coolaber, Beijing, China) was used. The CDS region of the *fxr* was constructed into pGBKT7 and pGBKT7-VP16 vectors, so that the experimental group of pGBKT7-FXR and pGBKT7-VP16-FXR plasmids was obtained. The experimental group plasmids and the blank plasmids (pGBKT7 and pGBKT7-VP16) were transformed into yeast Y2HGold competent cells, respectively, and then coated on SD/-Trp plate medium cultured at 28–30 °C for 2–3 days. Fresh single colonies were picked on each plate and cultured in SD/-Trp liquid medium. The OD600 was adjusted to 0.2 and diluted 10 times, 100 times and 1000 times with TE buffer (Sangon Biotech, Shanghai, China). The above bacterial solution (7 μL) was applied to the SD/-Trp, SD/-His/-Trp and SD/-Ade/-His/-Trp plates in order to be cultured at 28–30 °C for 2–3 days. The transcriptional activity of FXR was determined according to the growth of bacterial liquid on each plate.

### 2.5. Subcellular Localization

The FXR cDNA was homologously recombined with a pEASY-Basic Seamless Cloning and Assembly Kit (TRANS, Beijing, China) into the GFP-tagged pcDNA3.1 vector. The pcDNA3.1-EGFP as control and pcDNA3.1-FXR-EGFP plasmids were transfected, respectively, into liver primary cells using the universal transfection reagent (YEASEN, Shanghai, China), and then the subcellular localization of the target protein was determined under a fluorescence microscope (SEMPREX, Campbell, CA, USA).

### 2.6. Co-Immunoprecipitation (CO-IP)

The Human Embryonic Kidney 293 Cells (HEK293T) were purchased from the Cell Bank Culture Collection Center, Chinese Academy of Sciences. The HEK293T were cultured in the incubator (37 °C and 5% CO_2_) with medium including high-glucose DMEM (Solarbio, Beijing, China), 10% FBS and 1% penicillin and streptomycin. FXR and RXR were amplified by PCR with the homologous arm primers (Appendix A) of pcDNA3.1-Flag and pcDNA3.1-HA at the BamHI site, respectively, to construct pcDNA3.1-FXR-Flag and pcDNA3.1-RXR-HA plasmids. The plasmids were transfected into HEK293T cells. After 24 h, the medium was removed and the cell was digested with 0.25% trypsin. After that, the cell was washed twice with pre-cooled PBS, and centrifuged at 1000 rpm for 5 min. The cells were lysed with 200 μL IP lysate (Beyotime, Shanghai, China), including PMSF at 4 °C for 30 min, followed by centrifugation at 4 °C, 12,000× *g* for 10 min, and then the precipitate was discarded. The 30 μL lysate was taken as the input, and the remaining lysate was incubated overnight at 4 °C with Anti-HA magnetic beads (Beyotime, Shanghai, China). The 1 × protein loading buffer was added and boiled at 95 °C for 10 min. After cooling, the protein was placed on a magnetic frame for magnetic separation, and the supernatant was collected. SDS-PAGE gels were prepared according to the molecular weight of the target protein. Then, the same amount of protein was separated by electrophoresis under different treatments. The gel at the corresponding position was cut according to the molecular weight of the target protein and transferred to the polyvinylidene difluoride (PVDF) membrane activated by methanol in advance. The PVDF membrane was sealed in 5% skim milk at room temperature for 1–3 h. After blocking, the PVDF membrane was washed three times with TBST buffer, and the HA antibody (1:1000; CST, Boston USA) was diluted with Tris-buffered salineTween-20 (TBST) solution containing 1% skimmed milk powder. The PVDF membrane was placed in the corresponding antibody solution and incubated overnight in a chromatography freezer at 4 °C. After the HA antibody incubation, the PVDF strips were washed three times with TBST buffer at room temperature. The TBST solution of 1% skim milk powder was used to prepare the secondary antibody at a ratio of 1:5000. The PVDF membrane was incubated in the secondary antibody incubation solution for 1 h and washed three times with TBST. The resulting bands were visualized in a multifunctional gel imager (UVITEC, Cambridge, UK) using an ultra-sensitive ECL chemiluminescence kit (Beyotime, Shanghai, China).

### 2.7. Protein Molecular Docking

The 3D structure model of SWISSMODEL-FXR was used as the receptor protein, and SWISSMODEL- RXR was used as the ligand protein. After the cluspro program calculation, the cluster was sorted according to its size, which was selected as the optimal docking configuration, and then Pymol (https://pymol.org/2/) accessed on 23 December 2022 was used for 3D visualization analysis.

### 2.8. Western Blot (WB)

The total protein extraction method of turbot liver primary cells was as follows: After the cells in the six-well plate were treated with different glucose concentrations, the cells were washed three times with pre-cooled PBS, and 200 μL RIPA lysate (R0010, Solarbio, Beijing, China) containing 1% protease inhibitor (K0011, Beyotime, Shanghai, China) and phosphatase inhibitor (K0022, Beyotime, Shanghai, China) was added to each well. Subsequently, the cell lysate was ultrasonically shaken twice and centrifuged at 12000 rpm for 10 min of 4 °C to sample the supernatant. Nuclear and cytoplasmic protein extraction kits were used for extraction (P0027, Beyotime, Shanghai, China). The BCA protein concentration determination kit (P0012, Beyotime, Shanghai, China) was selected to determine the protein concentration. After adding 5 × protein loading buffer (P1040, Soleibao, Beijing, China), the proteins were heated at 95 °C for 5 min to denaturation, and the subsequent experimental steps were consistent with 2.6. The antibody used was as follows: β-ACTIN (1:50,000, AC026, ABclone, Wuhan, China), Lamin B1 (1:1000, WL01775, Vanke, China).

### 2.9. Immunofluorescence Analysis

The 24 mm× 24 mm coverslips (Biosharp, Beijing, China) were placed in the six-well plates (Corning, Lowell, MA, USA), and the liver cells of turbot were inoculated into the plate. After the cell confluence reached 70–80%, the medium was sucked out. The cells were fixed with 4% paraformaldehyde (Biosharp, Beijing, China) for 15 min, and then washed with PBS three times. The goat serum was added to the slide and blocked for 30 min. After sucking out the solution, a sufficient amount of diluted primary antibody (FXR: ABclonal, Wuhan, China; SHP: Omnimabs, Shanghai, China) was added to each slide and placed in a wet box to incubate at 4 °C overnight. The slides were washed three times with PBST for three minutes each time. After the excess liquid was removed with absorbent paper, the diluted fluorescent secondary antibody was added dropwise and incubated in a wet box at 20–37 °C for 1 h. The slides were washed three times with PBST for 3 min each time again. DAPI was added and incubated for 5 min in the dark. The specimens were stained and washed four times with PBST for 5 min each time. The excess liquid was sucked out with absorbent paper, and then the film was sealed with a sealing solution containing an anti-fluorescence quencher, and then the collected images were observed under a fluorescence microscope. Image J software V1.8.0.112 was selected to choose the red (green) fluorescence for area calculation, and then compared with the entire picture area, in which the obtained ratio was used as the relative expression of the protein (relative fluorescence area).

### 2.10. Enzyme-Linked Immunosorbent Assay (ELISA)

The kits used were double antibody one-step sandwich enzyme-linked immunosorbent assay kits (Yushao Biology, Shanghai, China). Glucose-6-phosphatase (G6PASE) and phosphoenolpyruvate carboxykinase (PEPCK) antibodies were pre-coated into the coated micropores. The specimens, standards and horseradish peroxidase-labeled (HRP) detection antibodies were added in turn. After incubation and thorough washing, tetramethylbenzidine (TMB) was converted into blue under the catalysis of peroxidase, and converted into the final yellow under the action of acid. The color depth was positively correlated with the antibody content in the sample. The absorbance (OD value) was measured with a microplate reader (Spectramax i3x, MolecularDevices, San Francisco, CA, USA) at a wavelength of 450 nm, and the activity of the sample was calculated.

### 2.11. Metabolite Detection

To detect the difference in metabolites in turbot hepatocytes, the cells were treated with different glucose concentrations and the supernatant was collected by centrifugation (1000 rpm, 10 min). The kits used for metabolite detection were provided by Nanjing Jiancheng Bioengineering Institute, China. Glucose and hepatic glucose output were detected by the glucose oxidase method (A154-1-1). The content of pyruvate was determined by colorimetry (A043-1-1), and the content of lactic acid was detected by enzyme catalysis (A019-2-1).

### 2.12. Dual-Luciferase Reporter Assay

The target gene was found on NCBI and the chromosome information of the gene was checked. After confirming the gene coding sequence, the promoter was selected to start the 3000 bp sequence. The promoter reporter plasmid was constructed by cloning and homologous recombination techniques, and confirmed by sequencing after successful construction. The PGL-reporter plasmid and CMV-expression plasmid were transfected according to the requirements of the luciferase reporter assay kit (Trans, Beijing, China), and the fluorescence value was finally read by the microplate reader. The transcription factor binding sites of *fxr* were predicted by the online website (https://jaspar.elixir.no/matrix/MA1110.1/) accessed on 6 January 2023. Through linearization, three sequences probably containing binding sites (Appendix A) were knocked out, respectively. After that, the corresponding PGL-reporter plasmid was constructed and the above experiments were carried out again.

### 2.13. Quantitative Real-Time PCR (qPCR)

The methods of total RNA extraction, quality detection and cDNA synthesis were consistent with those described in gene cloning. After that, first strand cDNA was diluted to 250 ng/μL using sterilized double-distilled water (Sangon Biotech, Shanghai, China). qPCR was carried out in a quantitative thermal cycle (Mastercycler^®^ eprealplex; Eppendorf, Hamburg, Germany). The amplification was performed in a total volume of 15 μL, containing 0.3 μL of each primer (10 mM), 4 μL of the diluted first strand cDNA product, 7.5 μL of 2× SYBR^®^ Premix Ex Taq™ II (Accurate Biology, Changsha, China) and 7.5 μL of sterilized double-distilled water. The qPCR program was conducted as follows: 95 °C for 2 min, followed by 40 cycles of 95 °C for 10 s, 58 °C for 10 s and 72 °C for 20 s. The gene expressions were determined using the 2^−ΔΔCt^. Housekeeping genes (*β-actin* and *rps4*) for turbot were selected from a pool of ten candidate housekeeping genes. All the primers used in the present study are listed in Appendix A.

### 2.14. Statistical Analysis

All statistical data were analyzed using SPSS 25.0 (IBM, Armonk, NY, USA). The normality of data distribution and the homogeneity of variance were tested. Then, one-way ANOVA and Turkey’s multiple range test were used for statistical analysis of gene expression among different groups. An independent-sample t-test was used to compare the difference between two groups. The data are presented in the form of mean ± standard deviation (mean ± SD), and the difference was significant at *p* < 0.05. OriginPro 9.1 software and Adobe Illustrator 2020 were used to create graphs and diagrams.

## 3. Results

### 3.1. High Glucose Can Affect the Expression of the FXR Gene and Protein in Turbot

The expression of FXR and SHP protein was detected via immunofluorescence after turbot hepatocytes were treated with different glucose concentrations for 24 h. Based on the fluorescence image (Figure 1A), the fluorescence intensity of FXR and SHP in the HG group (50 mM) was significantly weakened compared with that in the CON group (15 mM). The fluorescence region was quantified with Image J software (Figure 1B), and the results showed that the expression levels of FXR and SHP in the HG group were significantly lower than those in the CON group (*p* < 0.01). Although there was no significant difference, the gene expression of *fxr* and *shp* in the HG group showed a downward trend compared with that in the CON group (*p* > 0.05) (Figure 1C). Compared with those in the CON group, the total protein, cytoplasmic protein and nuclear protein of FXR were significantly decreased in the HG group (*p* < 0.05) (Figure 1D,E).

### 3.2. Gene Cloning and Analysis of fxr in Turbot

The full-length CDS region of *fxr* in turbot was 1458 bp, encoding 485 amino acids with the predicted molecular weight of 55.48 kDa, and the iso-site of 5.96. The deduced protein sequence of *fxr* displayed typical structures of a DNA binding domain (solid line region: 134–217) composed of two C4-type zinc finger structures, and a ligand binding domain (dotted line region: 261–480) (Figure 2A). The phylogenetic tree was constructed with the *fxr* in various fish species, reptiles, birds, mammals and amphibians, which showed that the most similar species to turbot were *Solea solea* and *Hippoglossus hippoglossus*, with the amino acid similarities both higher than 94% (Figure 2B).

### 3.3. Different Tissue Distribution of fxr in Turbot

The results of qPCR showed that *fxr* was detected in twelve tissues (skin, liver, intestine, muscle, gill, brain, spleen, stomach, adipose, head kidney, eye and heart) (Figure 3). Among them, the expression level of *fxr* was highest in the liver, followed by the heart tissue, which was dozens of times higher than in other tissues. Additionally, the expression of *fxr* was relatively high in adipose, eye and gill tissues. The gene expression of *fxr* was similar in the brain, head kidney, stomach, spleen, muscle and skin. The lowest expression of *fxr* was observed in the intestine (*p* < 0.05).

### 3.4. The Function of fxr as a Nuclear Transcription Factor Was Verified

The results of subcellular localization showed that the fluorescence was detected in the whole cell as the pcDNA3.1-EGFP plasmid was transfected into turbot liver cells, while the fluorescence was only observed in the nucleus when the pcDNA3.1-FXR-EGFP plasmid was transfected (Figure 4A). As shown in Figure 4B, the expression of FLAG protein was successfully detected by co-immunoprecipitation of HA-tagged proteins in the co-transfection system, indicating the protein interaction between FXR and RXR proteins. The results of the molecular docking prediction showed the spatial model of FXR and RXR forming heterodimers, and showed that RXR could bind to 15 binding sites of FXR by forming hydrogen bonds (Figure 5). The results of the dual-luciferase reporter assay showed that the activity of the SHP promoter was significantly up-regulated by co-transfection of the FXR plasmid and SHP promoter reporter plasmid compared with transfection of the SHP promoter reporter plasmid alone (*p* < 0.05). In addition, when co-transfected with FXR and its heterodimer RXR plasmids, the effect on the SHP promoter was enhanced by three times (*p* < 0.05) (Figure 4C).

To verify the transcriptional activity of FXR, the yeast transcriptional activity experiment was carried out. As shown in Figure 6, the four plasmids grew normally on the medium plate SD/-Trp, and the yeast growth gradually became sparse with the decrease in the OD value. On the SD/-His/-Trp plate, the growth activity of pGBKT7 was significantly decreased, while the growth ability of pGBKT7-FXR was significantly increased. Compared with pGBKT7-VP16, the growth ability of pGBKT7-VP16-FXR was significantly weakened. Similar growth results were observed on SD/-Ade/-His/-Trp plates, indicating that FXR has both transcriptional activation and transcriptional inhibition effects. 

### 3.5. FXR Activation Can Affect Glucose Metabolism in Turbot

Firstly, the activation levels of FXR endogenous agonist CDCA and exogenous agonist GW4064 were determined, as shown in Figure 7. Different concentrations of CDCA and GW4064 both had no significant difference in terms of the viability of turbot liver cells (*p* > 0.05) (Figure 7A). Based on the gene expression results of *fxr* and *shp* (Figure 7B), the activation level was the strongest when the concentration of CDCA was 50 μM and that of GW4064 was 2 μM, respectively (*p* < 0.05). After that, the activation levels of CDCA (0, 25 and 50 μM) and GW4064 (0, 1 and 2 μM) at different times were determined. The results showed that GW4064 with 2 μM had the best activation level at 24 h (Figure 7C), which would be chosen as the concentration and time treatment for subsequent activation experiments. In addition, the effects of activating *fxr* on glucose metabolism under different glucose concentrations were explored in Figure 8. In general, compared with that in the CON and HG groups, the expression of glycolysis-related genes (*gk* and *pk*) was significantly increased and that of gluconeogenesis-related genes (*cpepck*, *mpepck g6pase1*, *g6pase2* and *foxo1*) was significantly decreased in the CON-G and HG-G groups, and this effect was more significant in the HG-G group (*p* < 0.05).

The effects of different glucose concentrations activating FXR on hepatocyte metabolites are shown in Figure 9. Compared with that in the CON group, the concentration of glucose in the medium increased significantly in the HG group, while the condition of high glucose with activated FXR (HG-G) could significantly reduce the content of glucose in the medium (*p* < 0.05). There was no significant difference in pyruvate content between the groups (*p* > 0.05). Under the condition of normal glucose with activated FXR, the group (CON-G) showed significantly reduced lactate content in the medium compared with that in the CON group. High glucose treatment significantly reduced the content of lactate in the medium, while the lactate content in the HG-G group increased significantly (*p* < 0.05). Compared with that in the CON group, the HG group showed significantly increased hepatic glucose output, while the HG-G group showed significantly reduced glucose output of hepatocytes (*p* < 0.05) (Figure 9).

### 3.6. FXR Regulates the Expression of Downstream Genes by Binding to the Promoter

In order to explore the effect of turbot FXR on the promoter activity of glucose metabolism-related genes, a dual-luciferase reporter assay was performed (Figure 10). The results showed that co-transfection of FXR and RXR plasmids could significantly down-regulate the activity of the G6PASE2 promoter compared with only transfection of the G6PASE2 promoter reporter plasmid (*p* < 0.05) (Figure 10B). Compared with transfection of the cPEPCK promoter reporter plasmid alone, the activity of the cPEPCK promoter was significantly decreased with transfection of the turbot FXR plasmid (*p* < 0.05), and there was no significant difference compared with co-transfection RXR plasmid (*p* > 0.05) (Figure 10C). However, neither transfection of the FXR plasmid alone nor co-transfection with RXR plasmid could affect the promoter activity of mPEPCK, G6PASE1 or PK (*p* > 0.05) (Figure 10D–F).

To further explore the binding sites of FXR, three predicted binding sites were knocked out by linearization. As shown in Figure 11, after the removal of binding site A, whether it was transfected with FXR alone or co-transfected with FXR and RXR, there was no significant effect on the activity of the G6PASE2 promoter (*p* > 0.05). After removing the binding sites B and C, the activity of the G6PASE2 promoter was significantly decreased and the down-regulation effect was enhanced by almost twice when co-transfected with FXR and RXR plasmids (*p* < 0.05).

To determine the function of FXR in the gluconeogenesis pathway, the protein content of key enzymes G6PASE and PEPCK was detected (Figure 12). And the results showed that under the normal and high-glucose conditions, activated FXR significantly reduced the content of PEPCK in turbot hepatocytes (*p* < 0.05). Compared with that in the CON group, the content of G6PASE was significantly increased in the HG group, while under the condition of high glucose, activated FXR significantly reduced the content of G6PASE in cells only in the high-glucose group.

### 3.7. SHP Plays a Role in the Regulation of Glucose Metabolism Mediated by FXR

To explore the role of SHP in regulating glucose metabolism by FXR, SHP was knocked down in the primary hepatocyte of turbot under high-glucose conditions with activated FXR. The results showed that the siSHP-171 group had the highest knockdown efficiency (63%) (*p* < 0.05), and it was selected as the subsequent knockdown siRNA (Figure 13A). The expression of *gk* and *pk* in the HG-G-siSHP group decreased significantly compared with the HG-G group (*p* < 0.05). Compared with those in the CON group, gluconeogenesis-related genes (*cpepck*, *mpepck*, *g6pase1* and *g6pase2*) increased significantly in the HG group, and decreased significantly after activation of FXR (*p* < 0.05). The expression of *cpepck*, *mpepck* and *g6pase2* in the HG-G-siSHP group increased significantly compared with that in the HG-G group, while the expression of *foxo1* increased significantly or even exceeded that in the HG group (*p* < 0.05) (Figure 13B).

The SHP plasmid was overexpressed under high-glucose conditions to detect the expression of glucose metabolism-related genes (Figure 13C). Compared with the CON group, the HG-pcDNA3.1-SHP group had no significant effect on the glycolysis pathway, while it significantly reduced the expression of gluconeogenesis-related genes (*cpepck* and *g6pase2*). Compared with that in the HG-G group, the HG-pcDNA3.1-SHP group had a weaker ability to inhibit gluconeogenesis pathway-related genes, but significantly reduced the relative expression of *foxo1* (*p* < 0.05). By further transfection of the pcDNA3.1-SHP plasmid with different contents (0, 0.5, 2.5 and 5 ug), the results showed that the expression of *foxo1* gradually decreased with the increase in SHP concentration (*p* < 0.05), and reached the lowest value at the transfection level of 5 ug (Figure 13D).

## 4. Discussion

FXR has been confirmed to be involved in the regulation of glucose metabolism in mammals [13]. FXR in the liver of mice contributes to the coordinated regulation of the shift from hepatic glucose production to hepatic glucose utilization by interfering with carbohydrate-induced changes in gene expression [28]. The hepatic FXR/SHP axis regulates glucose/fatty acid homeostasis in aged mice [29]. However, the molecular pathway of FXR regulating glucose metabolism in fish remains unclear. Therefore, the expression of FXR and SHP was detected in high-glucose conditions to explore whether FXR is involved in glucose metabolism in turbot. High glucose significantly inhibited the expression of FXR and SHP genes and proteins in the liver, suggesting that FXR may be affected by glucose levels. A similar phenomenon was also found in diabetic rats, in which the expression of FXR in the liver was significantly decreased with the aggravation of diabetes [30]. FXR activity in mice has also been shown to be regulated by glucose flow in hepatocytes [31]. These results indicated that FXR in turbot could be affected by glucose homeostasis, which means its relationship with glucose metabolism was worthy of further exploration.

Accordingly, the gene cloning, mode of action and transcriptional function of FXR in turbot were first verified. In the present study, the CDs region of FXR was cloned and identified. It was found that the length of FXR CDs was 1458 bp, and encoded 485 amino acids, which had a typical DNA binding domain and ligand binding domain. The DNA binding domain of turbot was more conserved; therefore, it was speculated that FXR in turbot may have similar physiological regulation functions of glucose metabolism as in rats and human hepatocytes [13,29]. Subcellular localization can accurately locate the specific location of biological macromolecules in cells, such as in the nucleus, in the cytoplasm or on the cell membrane [32,33]. According to the subcellular localization, it was found that pcDNA3.1-FXR-EGFP was only expressed in the nucleus, which indicated FXR may have a nuclear localization ability. Subsequent yeast transcriptional activity experiments showed that pGBKT7-FXR could grow normally on both SD/His/-Trp and SD/His/Ade/-Trp defect media, while the growth of pGBKT7-FXR-VP16 on the two defective media was significantly limited. This suggests that FXR in turbot has both transcriptional activation and inhibition functions. In HepG2 cells, FXR activated by GW4064 suppressed apolipoprotein A-I transcription via a negative FXRE, which also confirmed the inhibition function of FXR [34]. Furthermore, combined with the following dual-luciferase reporter assay, FXR was confirmed to be a typical nuclear transcription factor with transcriptional activation and inhibition.

The highest expression of FXR in turbot was observed in the liver, which was consistent with the results in humans, mice and large yellow croaker [22,35]. The liver, as a metabolic center, was also the major place for FXR to regulate glucose homeostasis [12,36]. Therefore, the primary hepatocyte of turbot was selected to study the glucose metabolism mediated by FXR in the present study.

As a transcription factor, FXR can bind to the promoter of downstream genes alone or form heterodimers with retinoid X receptor (RXR) to regulate downstream genes involved in various physiological processes such as bile acid, glucose and lipid metabolism [34,37]. Through the co-immunoprecipitation analysis, FXR was confirmed to interact with RXR in turbot. Furthermore, two RXR splice variants, RXR5 and RXR6, were detected in the liver of turbot, and both of them could interact with FXR. Molecular docking is a theoretical simulation method that mainly studies the interaction between molecules, predicting their binding mode and affinity [38]. To explore the binding mode between the two proteins, Cluspro was selected for molecular docking, and it was found that FXR and RXR could be combined by forming hydrogen bonds. These results highlight the important roles of RXR heterodimerization in the nuclear receptor signaling of FXR, implying the function of FXR may be in the single or heterodimer form with RXR in turbot. Small heterodimer partner (SHP), as a nuclear receptor lacking a DNA binding domain, is both a transcriptional repressor and a direct target of FXR in humans [29,39]. In the present study, the dual-luciferase reporter assay showed that FXR could enhance the activity of the SHP promoter, which was significantly enhanced when co-transcribed with the RXR plasmid. Studies in humans have found that FXR can regulate glucose metabolism through gluconeogenesis mediated by SHP [29]. In addition, the transcriptional activity of FXR was enhanced when it formed the heterodimeric complex with RXR [37], which indicated that the function of FXR was highly conserved in different species. Notably, under high-glucose conditions, the protein level of SHP was significantly decreased, a tendency which was consistent with FXR. The above results indicate that RXR could directly regulate the activity of the SHP promoter in the form of a heterodimer with FXR, and revealed the possibility of SHP as a direct target gene of FXR to participate in downstream regulation.

As a typical bile acid receptor, FXR can be activated by different bile acid ligands, including endogenous bile acid ligands and specific exogenous ligands [14,40,41]. Studies have shown that the ligand specificity of bile salts was obtained in the late evolution of vertebrates, and different fish species had obvious differences in their ligand selection. For example, FXR was found to be effectively activated by chenodeoxycholic acid and GW4064 in the liver of *Oryziaslatipes* [25], while CDCA and GW4064 could not effectively activate the FXR in *Leucoraja erinacea* [42]. Therefore, it is necessary to select the FXR agonists of turbot. The results showed that both endogenous agonist CDCA and specific agonist GW4064 could effectively activate FXR, and the activation efficiency of GW4064 was significantly better than that of CDCA. Finally, GW4064 (2 μM) was selected as an agonist to activate FXR in normal and high-glucose conditions. The CON-G and HG-G groups could increase the expression of glycolysis-related genes (*gk* and *pk*) and reduce the expression of key genes in the gluconeogenesis pathway (*cpepck*, *mpepck*, *g6pase1* and *g6pase2*), which also further confirmed that the activation of FXR could alleviate the glucose metabolic disturbance. Non-sugar substances such as pyruvate, lactate, glycerol and amino acids can be used as precursors to generate glucose through the gluconeogenesis pathway to supplement the energy consumption [43]. Lactate, as the product of the glycolysis pathway, is also one of the substrates of gluconeogenesis [44] In the present study, the lactate showed an increasing trend in the HG-G group; meanwhile, the glucose level and liver glucose output decreased. Meanwhile, the gluconeogenesis pathway was significantly inhibited and the glycolysis pathway was improved in the HG-G group, indicating the content change in lactate may be caused by the change in the FXR-mediated gluconeogenesis and glycolysis pathway.

How does FXR regulate glucose metabolism? Given the characteristics of its transcription factors, the possibility of FXR directly binding to downstream target genes was first analyzed. FXR normally binds to the FXR response element (FXRE) in the form of a monomer or dimer with RXR to regulate the expression of multiple target genes [21]. In rats, it has been confirmed that typical FXREs comprise inverted repeats of the classical AGGTCA nucleotide sequence [20]. According to the prediction results for FXR transcription factor binding sites in turbot, the FXR binding sites were predicted in the promoter of *cpepck*, *mpepck*, *g6pase1*, *g6pase2* and *pk*. Notably, the dual-luciferase reporter assay results showed that FXR could only bind to the *mpepck* and *g6pase2* promoters to down-regulate their activity. The findings revealed that FXR is involved in regulating the gluconeogenesis pathway in turbot in the form of transcription factors. Subsequently, based on the FXR binding sites predicted by the JASPAR website, three sequences in the promoter of *g6pase2* were linearized to cut, respectively. The results showed that the *g6pase2* promoter without ‘ATGACCT’ lost the ability to bind to FXR, while ‘AGGTCA’, which is common in mice, was not the FXR binding element of turbot [17]. Similarly, the binding site sequence ‘ATGACCT’ in the promoter of *mpepck* was also found, which more accurately proved that turbot FXR could down-regulate the transcription of target genes by directly binding to the promoters of the key gluconeogenesis genes *mpepck* and *g6pase2.* To more comprehensively determine the effect of FXR on the gluconeogenesis pathway, the PEPCK and G6PASE activities were detected by ELISA, which showed that the activation of FXR under high glucose significantly inhibited the expression of PEPCK and G6PASE, indicating that the gluconeogenesis pathway was significantly inhibited. Combined with the results of the transcriptional activity verification and glucose metabolism-related gene expression, it was suggested that FXR activated by ligand GW4064 could bind to *mpepck* and *g6pase2* promoters and inhibit their transcriptional activity under high-glucose conditions, thereby inhibiting gluconeogenesis.

As a key target gene of FXR regulating glucose in mice and rats [13,17], what role does SHP play in turbot? The above results indicated that the regulatory effect of FXR on glucose metabolism was more obvious under high-glucose conditions. Therefore, FXR was activated under high-glucose conditions and SHP was knocked down to explore the role of SHP in the regulation of glucose metabolism mediated by FXR. After knocking down SHP, the effect of FXR activation in high glucose on the expression of glycolysis-related genes was reduced, and the inhibition of gluconeogenesis-related genes was also significantly reduced, indicating that FXR may regulate glucose metabolism through SHP. Moreover, the gluconeogenesis pathway was also inhibited by the overexpression of SHP in high glucose, although the inhibition was weaker than that of FXR activation, indicating the function of SHP involved in regulating gluconeogenesis by FXR. The function of the FXR-SHP negative regulatory cascade in targeting gluconeogenesis has been proven in mice [13]. Simultaneously, bile acids inhibit the expression of gluconeogenic genes, including *g6pase* and *pepck*, in an SHP-dependent fashion [17]. In the present study, the agonist GW4064 was selected as the specific activator of FXR; therefore, the high expression of SHP after GW4064 treatment was regulated by FXR. That is to say, the present study proved that FXR could promote the expression of SHP by enhancing the promoter activity, thus inhibiting the gluconeogenesis pathway. Furthermore, SHP plasmid transfection experiments showed that the expression of *foxo1* gradually decreased with the increase in SHP plasmid content. FOXO1, as a transcription factor, plays a key role in regulating glucose and lipid metabolism [45]. In our previous study, it was found that knockdown of *foxo1* significantly inhibited the expression of gluconeogenesis-related genes (*cpepck* and *g6pase1*) and increased the expression of glycolysis-related genes (*pk* and *gk*) in primary hepatocytes of turbot [6]. Therefore, FXR could inhibit the expression of *foxo1* through activating *shp*, thereby inhibiting the gluconeogenesis pathway and promoting the occurrence of glycolysis; however, there was no significant effect on the glycogen synthesis pathway. In the present study, FXR’s involvement in regulating glucose metabolism in turbot was only verified in vitro. The molecular mechanism of exogenous FXR activation to ameliorate high-carbohydrate diet-induced glucose and lipid metabolism disorders is being investigated in vivo.

## 5. Conclusions

In summary, FXR in turbot is a typical nuclear transcription factor with nuclear localization ability, which can form heterodimers with RXR by hydrogen bonding to regulate the expression of downstream target genes. FXR can directly bind to the promoters of crucial gluconeogenesis genes (*mpepck* and *g6pase2*) to inhibit the gluconeogenesis pathway by reducing its transcriptional activity. In another way, FXR also can up-regulate the activity of the SHP promoter in the form of dimers with RXR, thereby promoting the glycolysis process and inhibiting the occurrence of gluconeogenesis via down-regulating the expression of *foxo1*. These findings indicate that FXR can be selected as a molecular target to inhibit the gluconeogenesis pathway and alleviate glucose disorders caused by high glucose (Figure 14).

## Figures and Tables

**Figure 1 cells-13-01949-f001:**
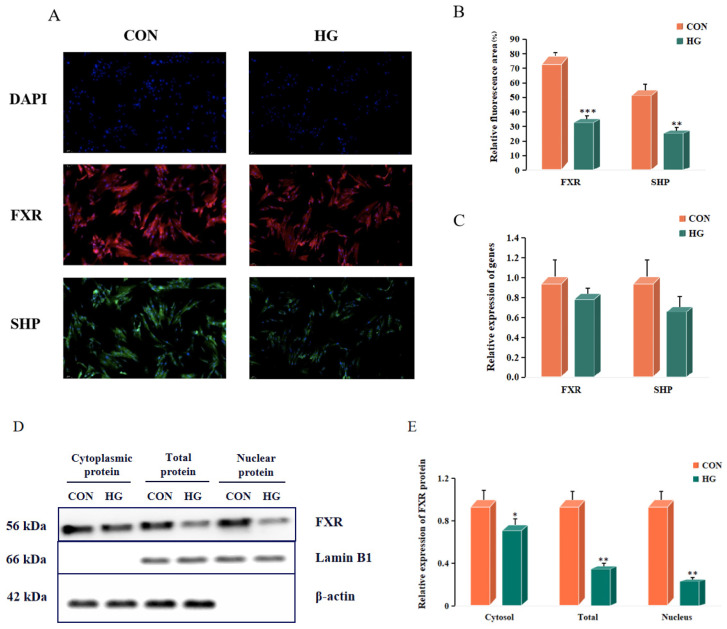
Gene and immunofluorescence expression of FXR and SHP in primary hepatocytes of turbot treated with different glucose levels. (**A**) Immunofluorescence results of FXR and SHP at different glucose levels. Red fluorescence represents the expression of FXR, green fluorescence represents the expression of SHP, and blue represents the nucleus stained with DAPI. (**B**) Quantitative map of immunofluorescence results of FXR and SHP. (**C**) Relative gene expression of *fxr* and *shp* at different glucose levels. (**D**) Relative expression of FXR protein in the cytosol, nucleus and total cell. (**E**) Quantitative map of FXR protein by gray value. CON: 15 mM, HG: 50 mM. The results are shown as means ± SD of 6 replicates. * *p* < 0.05, ** *p* < 0.01, *** *p* < 0.001.

**Figure 2 cells-13-01949-f002:**
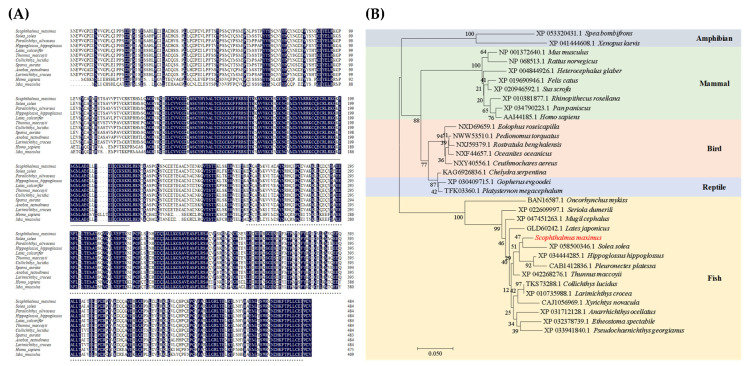
Deduced amino acid sequence and phylogenetic tree analysis of *fxr* in turbot. (**A**) Comparison of amino acid sequences of FXR. The solid line represents the DNA binding domain, and the dotted line represents the ligand binding domain. (**B**) The phylogenetic tree based on FXR amino acid sequences of turbot was constructed by the neighbor-joining method.

**Figure 3 cells-13-01949-f003:**
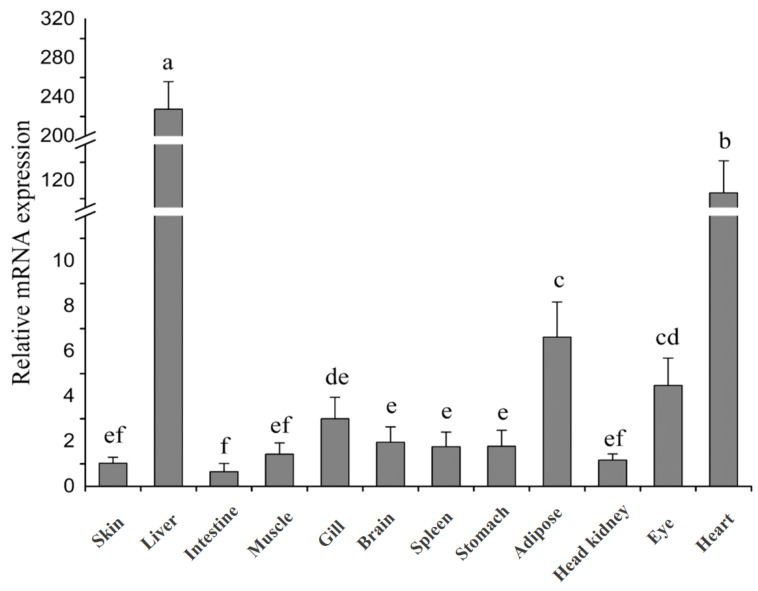
**Comparative analysis of *fxr* gene expression in different tissues of turbot.** The result is shown as means ± SD of 6 replicates. Different superscript letters indicate significant differences (*p* < 0.05).

**Figure 4 cells-13-01949-f004:**
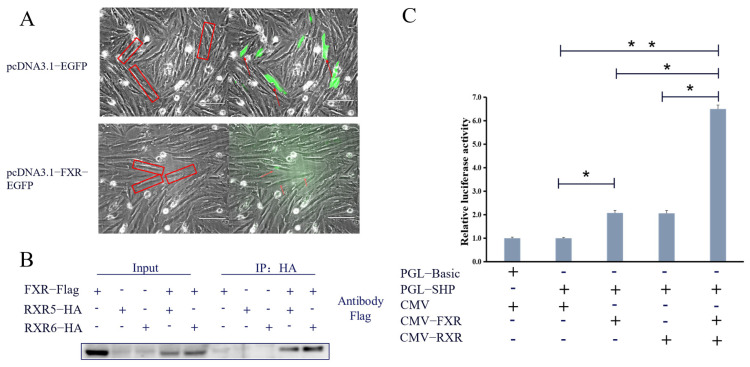
**Analysis of the FXR function with different methods.** (**A**) Subcellular localization of FXR in primary hepatic cells of turbot. The cells were observed through a fluorescence microscope; the left red frame corresponds to the right fluorescent cells. (**B**) The protein interaction between RXR and FXR by co-immunoprecipitation. (**C**) Effects of turbot FXR on SHP promoter activity by dual-luciferase activity analysis in HEK293 cells. * *p* < 0.05, ** *p* < 0.01.

**Figure 5 cells-13-01949-f005:**
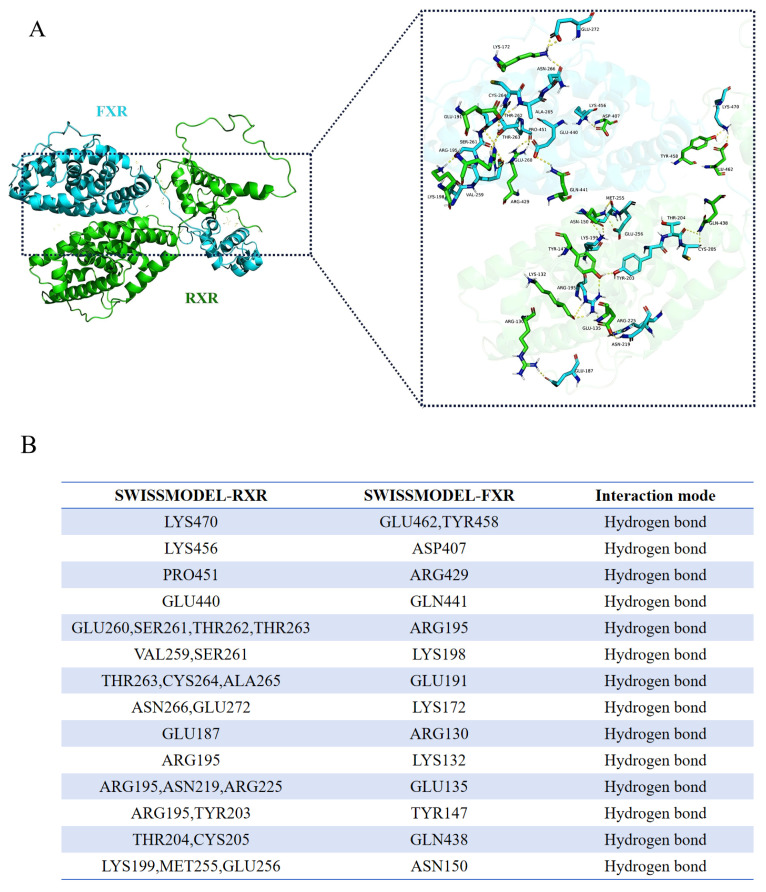
**Protein molecular docking model of FXR and RXR.** (**A**) The overall layout and local amplification of molecular docking of FXR and RXR; the yellow dotted line represents the hydrogen bond. (**B**) Binding sites of FXR and RXR.

**Figure 6 cells-13-01949-f006:**
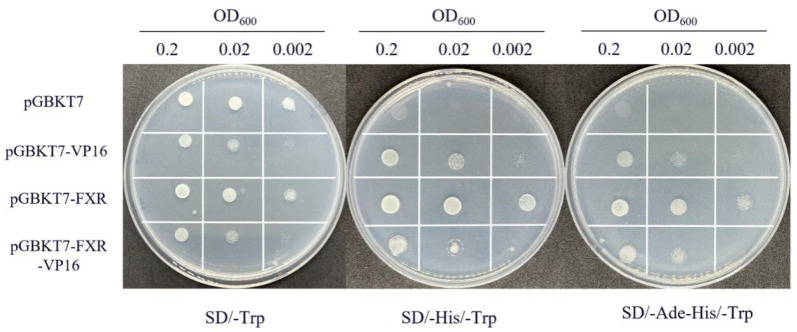
The transcription of downstream genes by FXR as a transcription factor.

**Figure 7 cells-13-01949-f007:**
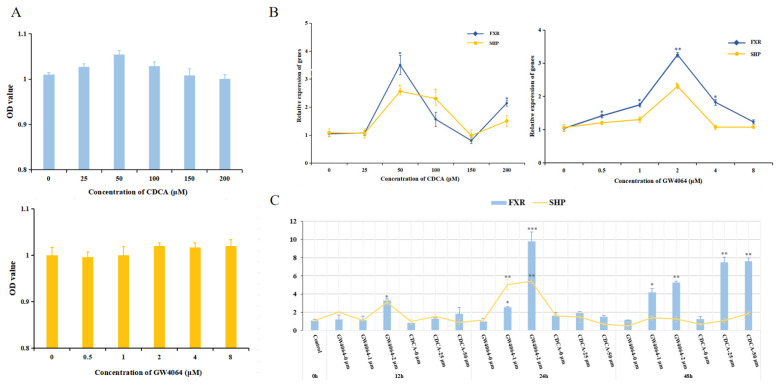
Comparison of the FXR activation levels between exogenous activator GW4064 and endogenous activator CDCA. (**A**) Effects of different concentrations of FXR activator on the viability of primary liver cells of turbot. CDCA: chenodeoxycholic acid; GW4064: FXR-specific agonist. (**B**) Concentration selection of CDCA and GW4064. (**C**) Treatment time point selection of different activators of FXR. The results are shown as means ± SD of 6 replicates. * *p* < 0.05, ** *p* < 0.01, *** *p* < 0.001.

**Figure 8 cells-13-01949-f008:**
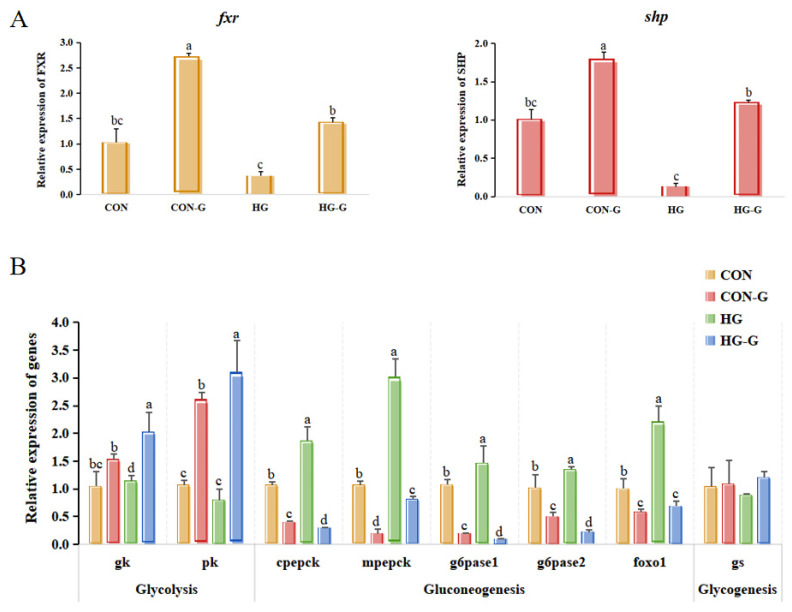
The effect of activating FXR on the expression of genes related to glucose metabolism under different glucose levels. (**A**) The effect of activators on activating FXR and SHP at two glucose levels. (**B**) The effect of activating FXR on the expression of genes related to glucose metabolism under different glucose levels. CON: 15 mM, CON-G: 15 mM + GW4064 (2 μM), HG: 50 mM, HG-G: 50 mM + GW4064 (2 μM). The results are shown as means ± SD of 6 replicates. Different superscript letters indicate significant differences (*p* < 0.05). glucokinase (*gk*); pyruvate kinase (*pk*); cytosolic phosphoenolpyruvate carboxykinase (*cpepck*); mitochondrial phosphoenolpyruvate carboxykinase (*mpepck*); glucose-6-phosphatase 1 (*g6pase1*); glucose-6-phosphatase 2 (*g6pase2*); glycogen synthetase (*gs*); forkhead box O1 (*foxo1*).

**Figure 9 cells-13-01949-f009:**
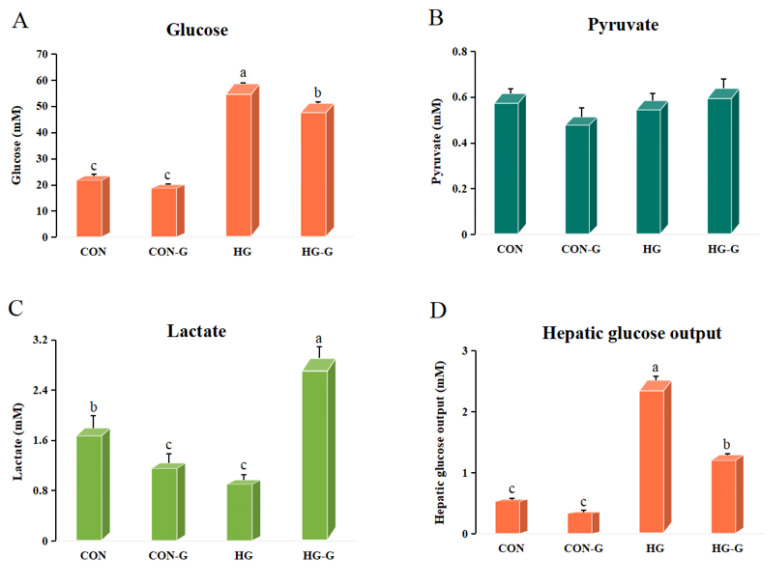
The effect of activating FXR on the hepatocyte metabolites under different glucose levels. (**A**) The effect of activating FXR on the glucose of hepatocyte under different glucose levels. (**B**) The effect of activating FXR on the pyruvate of hepatocyte under different glucose levels. (**C**) The effect of activating FXR on the lactate of hepatocyte under different glucose levels. (**D**) The effect of activating FXR on the hepatic glucose output of hepatocyte under different glucose levels. CON: 15 mM, CON-G: 15 mM + GW4064 (2 μM), HG: 50 mM, HG-G: 50 mM + GW4064 (2 μM). The results are shown as means ± SD of 6 replicates. Different superscript letters indicate significant differences (*p* < 0.05).

**Figure 10 cells-13-01949-f010:**
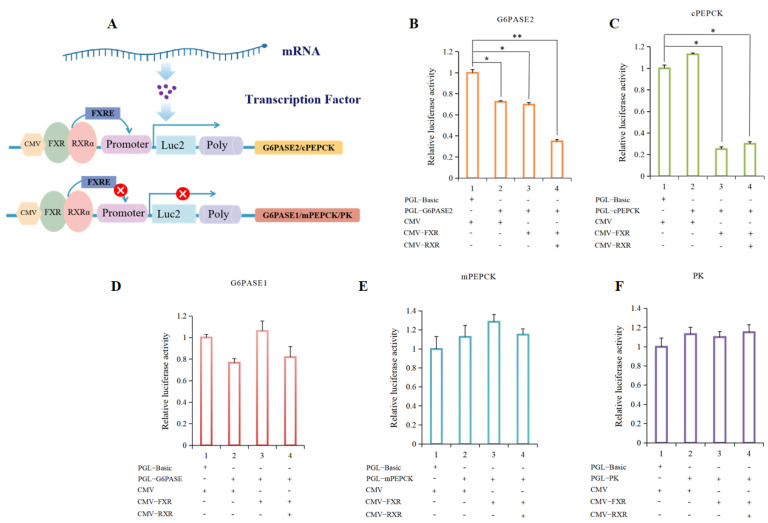
The effect of FXR on the promoter activity of glucose metabolism-related genes. (**A**) Schematic diagram of FXR as a transcription factor to activate the downstream promoter. And the red sign of “x” represents no effect of promoting promoter transcription (**B**) The results of FXR on the promoter activity of G6PASE2. (**C**) The results of FXR on the promoter activity of cPEPCK. (**D**) The results of FXR on the promoter activity of G6PASE1. (**E**) The results of FXR on the promoter activity of mPEPCK. (**F**) The results of FXR on the promoter activity of PK. The results are shown as means ± SD of 6 replicates. * *p* < 0.05, ** *p* < 0.01.

**Figure 11 cells-13-01949-f011:**
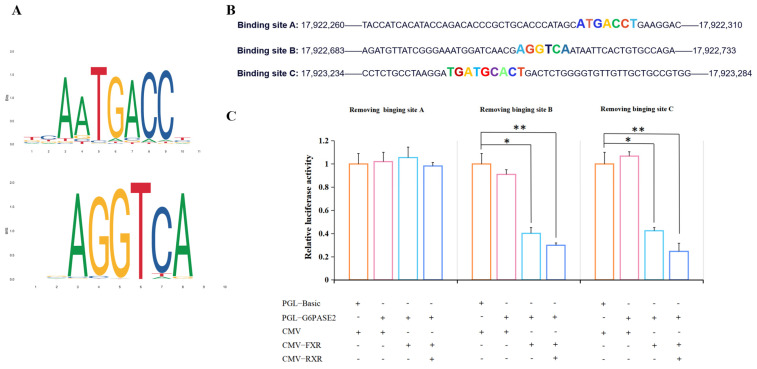
Prediction and confirmation of FXR binding sites. (**A**) The FXR binding site of turbot predicted by the website (https://jaspar.elixir.no/matrix/MA1110.1/) accessed on 6 June 2023. (**B**) Three binding sites of the G6PASE2 promoter selected for removal according to the prediction results. (**C**) The effect of FXR on the activity of the G6PASE2 promoter after deleting these three binding sites, respectively. The results are shown as means ± SD of 6 replicates. * *p* < 0.05, ** *p* < 0.01.

**Figure 12 cells-13-01949-f012:**
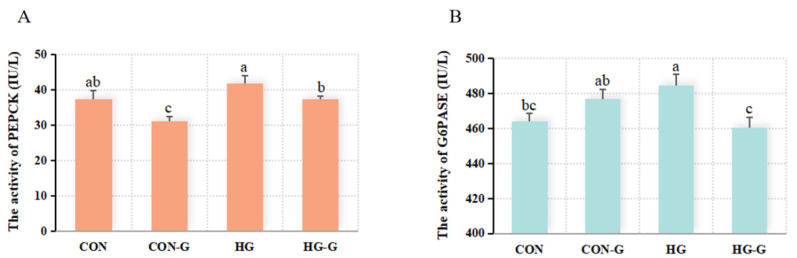
The effect of activating FXR on PEPCK and G6PASE under different glucose levels. (**A**) The result of activating FXR on PEPCK under different glucose levels. (**B**) The effect of activating FXR on G6PASE under different glucose levels. CON: 15 mM, CON-G: 15 mM + GW4064 (2 μM), HG: 50 mM, HG-G: 50 mM + GW4064 (2 μM). The results are shown as means ± SD of 6 replicates. Different superscript letters indicate significant differences (*p* < 0.05).

**Figure 13 cells-13-01949-f013:**
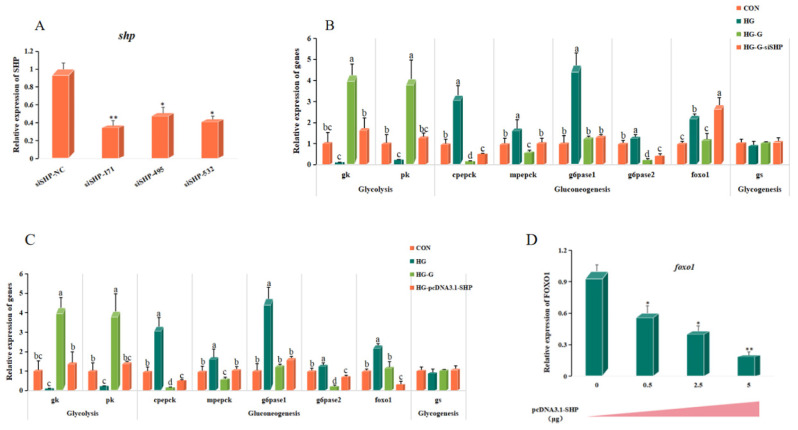
**The function of SHP in the FXR-mediated glucose metabolism pathway.** (**A**) The siRNA selection of *shp* through the expression of *shp*. (**B**) The effect on glucose metabolism of interfering SHP in high-glucose activated FXR. (**C**) The effect on glucose metabolism of overexpression of SHP in high-glucose conditions. (**D**) The foxo1 expression in transfection pcDNA3.1-SHP with different concentrations. CON: 15 mM, HG: 50 mM, HG-G: 50 mM + GW4064 (2 μM), HG-G-siSHP: 50 mM + GW4064 (2 μM) + siSHP, HG-pcDNA3.1-SHP: 50 mM + pcDNA3.1-SHP. The results are shown as means ± SD of 6 replicates. Different superscript letters indicate significant differences (*p* < 0.05). * *p* < 0.05, ** *p* < 0.01.

**Figure 14 cells-13-01949-f014:**
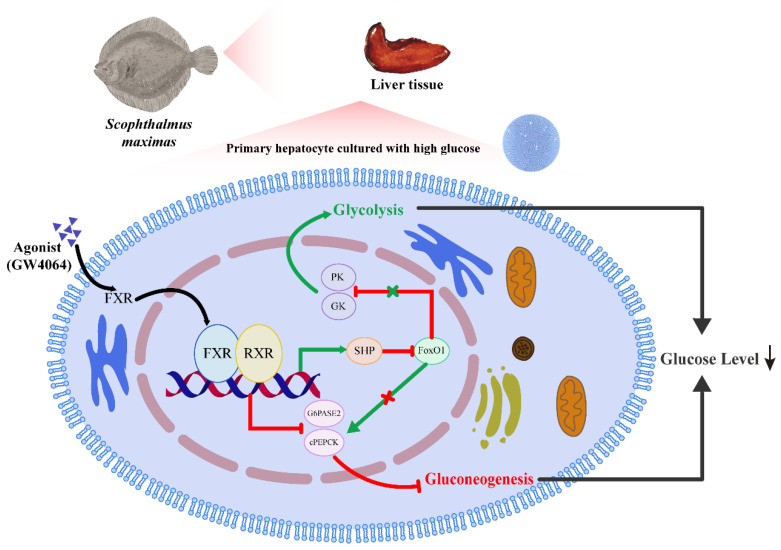
**The summary of the molecular pathway in the regulation of glucose metabolism by FXR.** GW4064 was selected to activate FXR in the primary hepatocyte cultured with high glucose. FXR could directly bind to the promoters of *mpepck* and *g6pase2* to reduce its transcriptional activity. FXR also could inhibit the expression of *foxo1* by activating the *shp* promoter, thereby promoting the glycolysis process and inhibiting the occurrence of gluconeogenesis, which played a role in regulating glucose homeostasis.

## Data Availability

Data is contained within the article or Appendix A.

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
