# Peer review of "The Molecular Mechanism of Farnesoid X Receptor Alleviating Glucose Intolerance in Turbot (Scophthalmus maximus)"

_cells, 2024, doi:10.3390/cells13231949_

Round 1
Reviewer 1 Report
Comments and Suggestions for Authors
The authors Gaochan Qin, et al. have submitted a Manuscript (ID: cells-3271183) in cells entitled “The molecular mechanism of farnesoid X receptor alleviating glucose intolerance in turbot Scophthalmus maximus”.
The authors used as model the fish Scophthalmus maximus to study the FXR gene and protein that was identified as a nuclear transcription factor.
Using, a great number of methods, was cloned the full length of FXR gene, analyzed the expression patterns, the promotor region, hepatocyte cell culture, knockdown of gene, qPCR, immunohistostaining, immunocytochemistry and bioinformatics analyses.
The result gives that the FXR had the highest expression in the liver and able to activate with SHP through heterodimer the expression of genes by binding to promoter, the date support that FXR of turbot has both transcriptional activation and inhibition functions.
I believe that all data of this study may be useful to increase the knowledge on that FXR role in gluconeogenesis pathway.
Finally, I consider the document in this form suitable, for publication in cells.
Author Response
Comments:
The authors Gaochan Qin, et al. have submitted a Manuscript (ID: cells-3271183) in cells entitled “The molecular mechanism of farnesoid X receptor alleviating glucose intolerance in turbot Scophthalmus maximus”.
The authors used as model the fish Scophthalmus maximus to study the FXR gene and protein that was identified as a nuclear transcription factor.
Using, a great number of methods, was cloned the full length of FXR gene, analyzed the expression patterns, the promotor region, hepatocyte cell culture, knockdown of gene, qPCR, immunohistostaining, immunocytochemistry and bioinformatics analyses.
The result gives that the FXR had the highest expression in the liver and able to activate with SHP through heterodimer the expression of genes by binding to promoter, the date support that FXR of turbot has both transcriptional activation and inhibition functions.
I believe that all data of this study may be useful to increase the knowledge on that FXR role in gluconeogenesis pathway.
Finally, I consider the document in this form suitable, for publication in cells.
Response: Thank you for your review and encouraging comments.
Reviewer 2 Report
Comments and Suggestions for Authors
MS presents very interesting results that focus on molecular mechanisms of glucose metabolism regulation in predatory fish, using turbot Scophthalmus maximus as an example. The main aim of MS was to investigate the role of Farnesoid X receptor (FXR), known to be a key player in glucose metabolism in mammals, but poorly understood in the context of fish. The results indicate that FXR, acting as a ligand-activated transcription factor, can inhibit gluconeogenesis in turbot hepatocytes, both by directly binding to gene promoters and indirectly by activating the SHP gene. It was observed that FXR activation reduces the expression of key gluconeogenic enzymes such as PEPCK and G6PASE, suggesting that FXR may be a potential molecular target in the regulation of glucose homeostasis in turbot. These results are important because they identify possible molecular pathways of glucose regulation, which may have applications in aquaculture and in studies on fish adaptation to different dietary conditions.
The study findings indicate that FXR can be selected as a molecular target to inhibit the gluconeogenesis pathway and alleviate glucose disorders caused by high glucose.
In my opinion, the MS is well planned and written. The methodology is well described. The results are very well presented and described. The positive value of the MS is the diagrams.
My minor comments can be found in the attached text.

Author Response
Comments 1: Remove keywords: Farnesoid X receptor; glucose intolerance; Scophthalmus maximus" and replace them with others, e.g.: amino acid sequences; gene expression; transcriptional activity of FXR
Response 1: Thanks for your suggestion, we have revised in the manuscript of page 3 and line 50 as suggested.
Comments 2: Was normality of distribution and homogeneity of variance tested? What program was used to create graphs and diagrams?
Response 2: Thank you for pointing this out. The normality of data distribution and the homogeneity of variance have been tested before statistical analysis. OriginPro 9.1 software and Adobe Illustrator 2020 were used to create graphs and diagrams. And the relevant information has been added to the manuscript (2.1.4) of page 16 and line 337.
Comments 3: Change the sentence to: The molecular mechanism of exogenous FXR activation to ameliorate high-carbohydrate diet-induced glucose and lipid metabolism disorders was investigated in vivo.
Response 3: Thanks, we have revised the manuscript of page 28 and line 623-625 as suggested.
Reviewer 3 Report
Comments and Suggestions for Authors
This article described the study to explore the molecular targets for regulating glucose metabolism in carnivorous fish, the turbot (Scophthalmus maximus). The paper is well-organized and written but you need to revise some parts.
1. There are many incorrect spellings throughout the menu script of this article. In particular, there is confusion about figure numbers, making it difficult for general readers to understand. For example, the following sentence on Page 9: "The results of molecular docking prediction showed the spatial model of FXR and RXR forming heterodimers, which found RXR could bind to 15 binding sites of FXR by forming hydrogen bonds (Figure 5)." By the way, it corresponds to Figure 6, not Figure 5. There are quite a few errors like this. You need to check.
2. Various figures in this paper also well prove the hypothesis. However, there are cases where there is only a figure and the explanation is not sufficient. For example, Figure 6 proves: "Protein molecular docking model of FXR and RXR" However, there is no detailed explanation about this. I recommend that you check this out and provide a detailed explanation.
3. In graphs including Figure 1, most results are expressed as relative values. An explanation is needed as to why. Additionally, in Figure 1B, the standard value of 100 is ambiguous. There is a need to clearly explain what the numbers are based on.
4. In Figure 2B, the authors conclude: "The most similar species with turbot were Solea solea and Hippoglossus hippoglossus, with the amino acid similarity both higher than 94%" However, among the species compared in Figure 1A, only H. hippoglossus is compared. I recommend adding comparative results for the remaining two species as well.
Author Response
Comments 1: There are many incorrect spellings throughout the menu script of this article. In particular, there is confusion about figure numbers, making it difficult for general readers to understand. For example, the following sentence on Page 9: "The results of molecular docking prediction showed the spatial model of FXR and RXR forming heterodimers, which found RXR could bind to 15 binding sites of FXR by forming hydrogen bonds (Figure 5)." By the way, it corresponds to Figure 6, not Figure 5. There are quite a few errors like this. You need to check.
Response 1: Thanks a lot, we have revised the incorrect spellings of the manuscript at page 18 (line 380, 387), page 19 (line 413) and l page 20 (line 435).
Comments 2: Various figures in this paper also well prove the hypothesis. However, there are cases where there is only a figure and the explanation is not sufficient. For example, Figure 6 proves: "Protein molecular docking model of FXR and RXR" However, there is no detailed explanation about this. I recommend that you check this out and provide a detailed explanation.
Response 2: As a heterodimer of FXR, RXR can bind to it to regulate the expression of downstream genes, which has been verified by CO-IP experiment (Figure 4B) and dual luciferase experiment (Figure 4C, Figure 10 and Figure 11). The molecular docking model of FXR and RXR was constructed to clarify the binding mode, and to highlight the important role of RXR heterodimer in FXR nuclear receptor signal transduction. In other words, the molecular docking model experiment is a supplement to the binding experiment of FXR and RXR.
Comments 3: In graphs including Figure 1, most results are expressed as relative values. An explanation is needed as to why. Additionally, in Figure 1B, the standard value of 100 is ambiguous. There is a need to clearly explain what the numbers are based on.
Response 3: Thanks a lot. The calculation unit (%) of Figure 1B has been added. We have supplemented the relevant calculation methods in the article of page 21 (line 280-283) and replaced the figure page 31 (Figure 1B) and made the following explanations about Figure 1.
The relative fluorescence area in Figure 1B means that the red (green) fluorescence area occupies the ratio of the entire picture. Image J software was selected to choose the red (green) for area calculation, and then compared with the entire picture area, therefore it was called the relative fluorescence area.
The relative gene expression in Figure 1C was detected by qRT-PCR. The CT values of the detected housekeeping genes (β-ACTIN and RPS4) were corrected for the difference of the CT values of the target genes (FXR and RXR). Therefore, the data obtained were relative gene expression.
About Figure 1E, the gray value of the protein was also quantified by Image J software, and then the relative protein expression was obtained by the calibration of the unified internal reference protein, which could reduce the error.
Comments 4: In Figure 2B, the authors conclude: "The most similar species with turbot were Solea solea and Hippoglossus hippoglossus, with the amino acid similarity both higher than 94%" However, among the species compared in Figure 1A, only H. hippoglossus is compared. I recommend adding comparative results for the remaining two species as well.
Response 4: Thank you for your insightful comment. The comparison of Solea solea with turbot in multiple sequence alignments have been added in Figure 1A according to your suggestion (page 32).
Round 2
Reviewer 3 Report
Comments and Suggestions for Authors
Your manuscript was reviewed carefully by myself and on the basis of this study, I have decided that the manuscript is acceptable in its modified manuscript.